# Restrained Shrinkage of High-Performance Ready-Mix Concrete Reinforced with Low Volume Fraction of Hybrid Fibers

**DOI:** 10.3390/polym14224934

**Published:** 2022-11-15

**Authors:** Hani Nassif, Mina Habib, Adi Obeidah, Mohammed Abed

**Affiliations:** 1Rutgers Infrastructure Monitoring and Evaluation (RIME) Group, Department of Civil and Environmental Engineering, Rutgers, The State University of New Jersey, Piscataway, NJ 08854, USA; 2Engineering Department, King’s College, Wilkes-Barre, PA 18702, USA

**Keywords:** hybrid fibers, high-performance concrete, restrained shrinkage, permeability, crack frequency

## Abstract

Cracking due to restrained shrinkage is a recurring issue with concrete bridge decks, impacting durability and ultimately service life. Several scholars’ research has proven that the incorporation of fibers in concrete mitigates restrained shrinkage cracking when utilizing high (0.5–3%) fiber volumes. This often presents a mixing and placement issue when used for ready-mixed concretes, which discourages their use in bridge decks. This study aims to optimize the incorporation of fibers for their benefits while producing concrete that is conducive to ready-mix, jobsite use. A series of tests were performed on a high-performance concrete (HPC) mix which incorporated blended, multiple fiber types (steel crimped, macro polypropylene, and micro polypropylene) while maintaining low total fiber (0.19–0.37%) volume. These “hybrid” fiber mixes were tested for multiple mechanical properties and durability aspects, with a focus on the AASHTO T334 ring test, to evaluate fiber efficiency under restrained conditions. Promising results indicate the use of a low-volume hybrid fiber addition, incorporating a macro and micro polypropylene fiber (0.35% by volume) blend, reduced the cracking area by 16.6% when compared to HPC incorporating a single fiber type, and 39% when compared to nonfibrous HPC control mixture.

## 1. Introduction

Structural concrete cracking due to restrained shrinkage is an ongoing issue with concrete bridges, impacting durability and service life. A common solution to mitigate restrained shrinkage cracking in concrete is the addition of randomly distributed fibers [1,2,3,4]. Experimental programs and studies concluded that introducing fibers into concrete is effective in reducing concrete brittleness, poor fracture toughness, and low tensile strength, as well as controlling crack propagation and reducing crack width [5,6,7]. Steel and synthetic fibers are two widely utilized fibers; they significantly enhance the stiffness and toughness of concrete [8,9]. Specific fiber types are effective only for a particular dimension of crack size, dependent on the material composition, aspect ratio, and modulus of elasticity [10]. Fiber hybridization, optimizing the addition of multiple types, is a remarkable development in concrete mix production. Concrete with attractive and improved engineering properties can be produced using the benefits of multiple types or sizes of fibers [11,12].

Rajarajeshwari et al. [13] found that hybridization of 1% steel fiber and 0.036% polypropylene fiber increased 28-days compressive and splitting tensile strength of concrete up to 17.11% and 52.87%, respectively. Teng et al. [14] investigated fiber hybridization’s effect on concrete flexural toughness, utilizing combinations of metallic and nonmetallic fibers. Their results show that increasing the metallic fiber percentage of overall hybrid fiber volume increases the flexural toughness and reduces the concrete’s resistivity to chloride permeability. Similar findings were published by Zhong et al. [15], who noted that the inclusion of steel fiber would increase the number of connected fibers, which could reduce the concrete’s resistance to chloride permeability. This can be mitigated by the incorporation of higher polypropylene fiber dosage with steel fiber, which will enhance the concrete’s engineering properties with no adverse effect on the concrete’s resistance to chloride permeability [15,16]. In addition to testing the chloride permeability of hybrid fibers, Afroughsabet et al. [16] observed that the hybridization of 0.3% polypropylene fiber and 0.7% steel fiber by volume reduces shrinkage strain by up to 26%, provides stabilization for the free shrinkage that occurs over time, and controls crack propagation.

Sun et al. [17] noted that incorporating hybrid fibers in high-performance concrete (HPC) resulted in fewer shrinkage cracks than utilizing a single fiber type. They determined that the hybridization of different sizes and types of fibers decreased permeability, inhibited the initiation and propagation of cracks, improved the pore structure of HPC, and decreased the shrinkage strain. Additionally, Sivakumar et al. [18] investigated the synergistic behavior of hybrid combinations of steel and nonmetallic fiber in controlling plastic shrinkage cracks using an advanced image analysis technique. The authors observed that the hybridization of steel and polypropylene fibers reduced the size and number of cracks, inhibiting shrinkage by reducing fiber spacing. Similarly, Cao et al. [19] tested the effect of hybridizing steel fibers, polyvinyl alcohol fibers, and cheap calcium carbonate whiskers on the plastic shrinkage of concrete. Their results showed an improvement in the cracking performance compared to the plain control mix. Even though many scholars confirmed that fibers could reduce concrete shrinkage, Al-Kamyani et al. [20] reported that the hybridization of fibers may increase air voids and potentially increase free shrinkage, but they noted that further research is needed in this area. The impact of fibers on concrete shrinkage, particularly mixes with low fiber content, is not clearly understood. Incorporating a minimal hybridized fiber proportion (up to 0.50% by volume) to high-strength concrete is effective in improving mechanical performance and strength at all testing ages [21], and further understanding of that low percentage of fibers will facilitate the production of fiber-reinforced concrete.

Controlling cracks induced by restrained shrinkage in structural applications, particularly bridges, is necessary; thus, bridge decks and components are a common application for fiber-reinforced concrete [22]. Most bridge decks are composite, consisting of a reinforced concrete slab with steel girders as support. The restrained shrinkage cracks in bridge decks commonly form where girders are located, as a result of the internal and external restraints. The external restraint is due to the sub-base or bridge superstructure, while the internal restraint is due to the reinforcement and aggregate [23,24]. Quantifying the degree of the imposed restraint is complex and dependent on multiple factors; however, several tests have been introduced to evaluate the restrained shrinkage cracking, with the AASHTO T334 ring test the most widely used [25,26,27,28]. Yıldırım [25] conducted ring testing on hybrid fiber concrete at 2% by volume and recognized that the steel and polypropylene fiber hybridization delayed the time to crack initiation. The incorporation of multiple fiber sizes, as compared to using a single type of fiber, reduces the total crack width measurements. Different types and sizes of fibers capture the different scales of microstructural flaws and therefore help to reduce restrained shrinkage.

Previous research largely focused on the use of hybrid fibers in concrete at a volume fraction between 0.5% and 3% [29], with very limited experimental data about the hybridization of fibers at a volume fraction less than 0.5%. This study aims to optimize low-volume hybrid fiber concrete and foster the use of such technology in ready-mix concrete, thereby expanding the application of fiber-reinforced concrete. The AASHTO T334 ring test [30], utilized to evaluate restrained shrinkage cracking, mechanical properties, and durability performance of hybrid fiber-reinforced high-performance concrete (HFHPC), is tested. Mixes were produced by incorporating varying hybrid blends of steel and synthetic fibers at a volume fraction between 0.19 and 0.37%. Thus, the potential of HFHPC to enhance the performance of concrete structures and improve cracking resistance is highlighted.

## 2. Materials and Methods

### 2.1. Materials

Producing HFHPC requires utilizing cement, supplementary cementitious materials, coarse aggregate, fine aggregate, fibers, water, and chemical admixtures to meet applicable ASTM standards. The cement utilized was Type I Portland Cement with supplementary cementitious materials incorporated to improve the mechanical performance of the concrete mixes [31]. The utilized supplementary cementitious materials were processed Class F fly ash and densified silica fume. Two aggregate sizes were used in this study: 19 and 9.5 mm. The specific gravity of those aggregates is 2.827 with an absorption capacity of 1.1%. The fine aggregate had a specific gravity of 2.618 and an absorption of 0.4%. Both fine and coarse aggregates were provided by Clayton Block Cooperation. Table 1 depicts the three types of fibers utilized in this study: macro polypropylene fiber (M), monofilament micro polypropylene fiber (N), and crimped steel fiber (S). These three types of fibers are commonly used in the production of HFHPC; however, the hybridization of two of them together in low volume fraction was investigated in this study.

### 2.2. Mix Proportions

Eight mixtures were prepared in the laboratory to analyze the effect of fiber hybridization on the performance of HFHPC. All mixtures included a blend of both #8 (9.5 mm) and #67 (19 mm) coarse aggregates in a ratio of one-to-five. Coarse aggregate blending enhances concrete pumpability and improves the fiber distribution throughout the concrete matrix. High-range water-reducing (HRWR) admixture was used to achieve the desired workability for each HFHPC mixture. The proportions and fiber dosages are summarized in Table 2 and Table 3.

#### 2.2.1. Control Concrete Mixtures

High-performance concrete (HPC): New Jersey Turnpike Authority standard HPC mix design [32]. The cementitious proportion consists of Type I cement, silica fume (3.7%), and Class F fly ash (20%).Steel fiber-reinforced high-performance concrete (S32): identical to HPC mix design with the addition of 25 kg/m^3^ of “S” fibers (0.32% by volume).Polypropylene fiber-reinforced high-performance concrete (M32): Identical to HPC mix design with the addition of 3 kg/m^3^ of “M” fibers (0.32% by volume). This mix was successfully pumped and implemented in reconstructed bridge decks in New Jersey [33].

#### 2.2.2. Hybrid Fiber-Reinforced Concrete Mixtures

M16N03: HPC mix design with the addition of 1.5 kg/m^3^ (0.16% by volume) of “M” fibers and 0.3 kg/m^3^ (0.03% by volume) of “N” fibers.M32N03: HPC mix design with the addition of 3 kg/m^3^ (0.32% by volume) of “M” fibers and 0.3 kg/m^3^ (0.03% by volume) of “N” fibers.N03S10: HPC mix design with the addition of 7.4 kg/m^3^ (0.1% by volume) of “S” fibers and 0.3 kg/m^3^ (0.03% by volume) of “N” fibers.M32S04: HPC mix design with the addition of 3 kg/m^3^ (0.32% by volume) of “M” fibers and 3 kg/m^3^ (0.04% by volume) of “S” fibers.M32S05: HPC mix design with the addition of 3 kg/m^3^ (0.32% by volume) of “M” fibers and 4 kg/m^3^ (0.05% by volume) of “S” fibers.

### 2.3. Mixing, Curing, and Testing

The eight mixtures were prepared in a rotating drum mixer, utilizing a modified ASTM C192 method [34]. Fibers were gradually added to each mixture to prevent clumping, and the mixing time was extended by 3 min. This modification ensured that the supplementary cementitious materials, chemical admixtures, and fibers were well mixed [35]. Fresh properties were tested and the samples (cylinders, prisms, and AASHTO T334 shrinkage rings) were then cast appropriately and placed in an environmental chamber. For mechanical priorities testing, three samples were cast to be tested at each age. After 24 h of initial curing in the environmental chamber, all samples were demolded, wet-cured for 14 days, then kept in laboratory air storage until completion of testing. This modified curing period was implemented to mimic field wet curing, based on a study conducted by Nassif and Suksawang [36] which concluded that a minimum of 14 days of wet curing is required for HPC to attain its durability and strength performances.

For evaluating HFHPC’s fresh properties, slump and air content were tested, and for evaluating its mechanical properties, compressive strength, tensile strength, elastic modulus, and modulus of rupture tests were conducted. Surface resistivity, rapid chloride permeability, free shrinkage, and restrained shrinkage were tested as indicators of durability properties. All testing was in accordance with ASTM and AASHTO standards to the age of 56 days and is summarized in Table 4.

The AASHTO T334 ring test [30] was the primary test utilized to measure and quantify the cracking of the trial mixes. The test consists of an inner steel ring with four foil strain gauges attached to its inner surface, with an concrete ring cast around the outer surface of the steel ring. The dimensions of the ring are illustrated in Figure 1a. The inner steel ring restrains the concrete from shrinking, and as a result, a hoop tensile stress is induced in the concrete test sample. If this stress exceeds the tensile strength of the mix, the concrete begins cracking. The AASHTO T334 standard states that any mix that does not develop a full crack within the testing period is deemed adequate for restrained shrinkage applications.

## 3. Testing Results

### 3.1. Fresh Properties

The test data compiled in Table 5 illustrate that the addition of fibers negatively affected concrete workability, as many researchers have previously observed [37]. Increased dosages of HRWR admixture were required to counteract the loss of workability. Incorporation of micro or macro polypropylene fiber into HPC required a significantly higher dosage of HRWR, reaching 21% in the case of M16N03. On the other hand, when steel fiber is utilized, the increase in HRWR dosage did not exceed 10%.

### 3.2. Mechanical Properties

#### 3.2.1. Compressive Strength

Figure 2a indicates that adding fibers tends to slightly increase the compressive strength of HFHPC, particularly at early ages. Comparing the HPC control mix with the series of HFHPC mixes showed improved compressive strength throughout the 1-, 7-, and 14-day timeframes. The increase for mix M32N03 was 27.4% at 1 day and 15.4% at 7 days when compared to the control mix. Improvement was also observed with mixes N03S10 and M32S4; both displayed a 19% increase at 1 day and 10% at 14 days. The improved compressive strength of HFHPC becomes less pronounced at later ages. M16N03 and M32S04 both improved their 28-day compressive strengths by 3.3% as compared to the HPC control mix, and up to 4.3% compared to S32 and M32 mixtures. Similarly, M32N03′s strength at 28 days was 6.4% higher than the HPC control mix and 7.5% higher than both S32 and M32 mixes. Although the comparative improvement of the fiber-reinforced mix strengths became moderated at later ages, the use of fibers assisted in attaining 69–75% of the 28-days strength at 7 days, while the control only reached 63%.

#### 3.2.2. Splitting Tensile Strength

Figure 2b shows the splitting tensile results. The results show that the incorporation of a single fiber type increased splitting tensile strength by up to 27% and 14% at 1 and 7 days, respectively, when compared to the HPC control mix. When hybrid fibers are added, tensile capacity increased up to 27.6% at 1 day and 25% at 7 days, compared to the control. By 28 days, the M16N03, M32N03, and M32S04 fiber mixes show significant improvement in the splitting tensile.

When comparing hybrid fiber mixes M16N03, M32N03, and M32S04 to single fiber control mixes S32 and M32 at 28 days, splitting tensile strength increased up to 10% and 4%, respectively. At 56 days, M16N03 splitting tensile strength is 18.6%, 18.2%, and 9.6% higher when compared to the HPC, S32, and M32 control mixes, respectively. These results confirm the significant positive effect of hybridizing, especially with low total polypropylene and steel fiber content (lower than 1.0%), as investigated by Qian and Stroeven [38]. Hybrid combinations of macro and micro polypropylene fibers provide higher resistance to tension when compared to macro polypropylene fiber alone. This is attributed to the synergistic effect created when macro fibers are combined with micro polypropylene fibers. Moreover, the use of fibers assisted in attaining 80–93% of the 28-days strength at 7 days, compared to only 75% for the HPC control mix.

#### 3.2.3. Modulus of Elasticity

All mixes with fibers improved the elastic modulus by 5% when compared to the HPC control mix, which exhibited only a slight change in elastic modulus at 28 days. However, it can be concluded that the effect of fiber hybridization on the elastic modulus of HFHPC is not significant, unlike its splitting tensile strength.

#### 3.2.4. Modulus of Rupture

Figure 3a graphs the modulus of rupture test data, showing that all hybrid fiber-reinforced mixes improved flexural strength from 12.4% (M32S5, M32N03) to 24.5% (M32S4) at 28 days when compared to the HPC control (no fiber) mix. Figure 3b presents the increase (%) in modulus of rupture achieved by hybridized fiber mixes as compared to the HPC, S32, and M32 control mixes. Hybridizing either micro or macro polypropylene with steel fibers (N03S10, M32S4) resulted in the greatest improvements in modulus of rupture. N03S10 (0.13% by vol.) flexural capacity improved by 24.5%, 23.1%, and 14.7%, as compared to HPC, S32, and M32 control mixes, respectively. Moreover, M32S4 modulus of rupture, when compared to the control mixes, improved by 19.7% compared to HPC, was 18.4% greater than the S32 mix, and 10.22% higher than the M32 mix. Improvements were also observed with M16N03, M32N03, and M32S05 mixes: up to ~12% when compared to the HPC control mix, ~11% as compared to S32, and ~3.5% as compared to M32. The findings of this study concur with several other studies [38,39]. Akcay and Tasdemir [37] stated that improving the fracture energy and ductility ratios of concrete are the most significant effects of utilizing the fiber hybridization.

### 3.3. Durability Properties

Surface resistivity and rapid chloride permeability tests were conducted to evaluate the durability properties of HFHPC, as shown in Figure 4. Comparing mixes, the graphs illustrate the effect of adding and combining fibers on the rapid chloride permeability and surface resistivity testing of the HFHPC. The nonfibrous HPC control mix performs the best in terms of rapid chloride permeability and surface resistivity, as expected due to the curing rates of HPC and the use of fly ash [40]. The results confirm that the addition of fibers affected both resistivity and permeability, as expected. The graphs show that steel fiber increased the readings significantly, from 85% for (S32) to 153% for (M32S5), which increases the danger of corrosion rate for the reinforcement of those mixes. On the other hand, this adverse effect on permeability was observed to be less significant when using macro and/or micro polypropylene fibers (2.5–41%).

When comparing M32 results to M32S4 and M32S5, the permeability was adversely affected by +110% and +134% with the addition of 0.04% or 0.05% (by volume) of steel fiber to the mix. However, M32′s permeability did not change significantly when combined with microfiber mix M32N03. The results presented are similar to the results of a study conducted by Feng et al. [41], which investigated the microstructure of fiber-reinforced concrete produced by either steel or polypropylene fibers. They concluded that polypropylene fiber creates a stronger interfacial transition zone within a concrete matrix, explaining the low permeability of the fiber-reinforced concrete produced by polypropylene fiber as compared to the fiber-reinforced concrete produced with steel fibers. For reference, Table 6 presents the surface resistivity and rapid chloride permeability limits specified by AASHTO TP 95 and ASTM C1202. Comparing these limits with the results of this study, all trial mixtures are categorized as “Low” or Very Low” permeability concretes.

Figure 5 illustrates the correlation between the chloride permeability and surface resistivity results. The correlation attempts to predict the rapid chloride permeability of the HFHPC at 56 days by using the surface resistivity measurements at 7, 14, and 28 days. This correlation could assist transportation agencies to approve mixes prior to 56 days, based on the surface resistivity measurements.

### 3.4. Free Shrinkage

Three free shrinkage tests were prepared as per ASTM C157 (75 × 75 × 285 mm prisms) and tested for each mix, with average values reported. As presented in Figure 6, the HPC control mix experienced significantly higher shrinkage than all other mixes. Among all mixes, S32 and M32N03 exhibited the lowest free shrinkage strain, an improvement of up to 18.5% and 22% as compared to the control mix. M32 exhibited the least shrinkage improvement at 1.8%; however, results do show that when macro polypropylene fiber is combined with micro polypropylene or steel fibers, shrinkage decreases significantly. The graph illustrates that the addition of micro polypropylene fibers to M32 resulted in lowering free shrinkage up to 21% (M32N03) and the addition of steel fibers (M32S5) resulted in an 11% improvement. In general, utilizing fiber hybridization reduces the free shrinkage of HPC, with the rate of improvement depending on the fiber types utilized.

## 4. Restrained Shrinkage Results

Throughout the duration of the study, cracks were visually inspected and measured. Since all the mixes in the present study did not develop full-depth cracks, analysis of the microcracks became a critical part of this experiment. Microcrack inspection was performed with the use of a portable microscopic camera, as shown in Figure 1b. The rings were scanned across the outer surface and the number of cracks was determined. Typically, if the stress in the concrete is increasing, the crack width will also increase. Table 7 and Figure 7 summarize the test results of the test rings, including the number of cracks, maximum crack width, and cracking area, at the ages of 28 and 56 days.

### 4.1. Number of Cracks

Figure 7a quantifies the number of cracks resulting from all concrete mixes. The HPC control mix exhibited the highest number of cracks at 56 days. When compared with the HPC control mix at 56 days, S32 and M32 reduced the number of cracks by 19.3% and 16.8%. When micro and macro polypropylene fibers are combined (M32N03) and compared to M32 at 56 days, the number of cracks was further reduced by 14.6%. In addition, by adding steel fiber to macro polypropylene fiber, mixes M32S4 and M32S5 were also successful in reducing the number of cracks up to 10.7% and 16.7%, respectively, when compared to M32.

### 4.2. Maximum Crack Width

Figure 7b illustrates that mixes M32 and M32N03 achieved the same maximum crack width at 56 days, an improvement of 33.3% when compared to the HPC control mix. In addition, M32N03, M32S04, and M32S05 outperform S32 by 8.7%, 2.9%, and 5.8%, respectively. However, they exhibited a greater maximum crack width than M32. All HFHPC mixes produced maximum crack widths that did not exceed commonly established maximum allowable crack widths. AASHTO and ACI 318 codes state that the maximum allowable crack widths under severe environmental exposure and at the tensile face of reinforced concrete structures are 0.3 mm and 0.33 mm, respectively [25,30]. None of the HFHPC mixes produced for this study exceeded these limits, and fiber hybridization aided in reducing the maximum crack width, decreasing the number of cracks, and delaying the crack initiation time.

### 4.3. Cracking Area

The cracking area is a parameter that is significantly improved with the addition of any fiber type. The area is calculated by summing the average width by the length of each crack. Figure 7c illustrates that all hybrid mixes outperformed the HPC control mix. At 56 days, M32N03 achieved 16.6% and 15.7% lower cracking area when compared to both M32 and S32. When compared to S32, both M32S04 and M32S05 reduce the cracking area by 10.2% and 9.7%, respectively. They also reduce the cracking area in comparison to M32 by 9.2% and 8.7%, respectively. Hybrid fibers containing macro and micro polypropylene fibers performed best concerning the cracking area; thus, the restrained shrinkage induction was decreased, representing lower concrete stress.

### 4.4. Strain Measurement

Figure 8a provides a graphical representation of the average strain data obtained from the foil strain gages attached to the steel ring. Overall, the incorporation of fibers results in a much lower strain level than the HPC (nonfibrous) control mix. At 7 days, the hybrid mixtures with micro polypropylene fibers (M32S5, M32N03, M16N03) achieved a 42% lower strain when compared to the control mix and reduced by 37% as compared to M32. Most fiber-reinforced mixtures at 56 days presented similar performance, the exceptions being M32S04 (−13%) and M32S05 (−39%), as illustrated in Figure 8b. Restrained shrinkage induction was decreased by utilizing fiber hybridization in the HPC, representing lower concrete stress. The data are shown in Figure 8 and, much as the Figure 7c cracking area data do, indicate that hybrid fiber mixes containing macro and micro polypropylene fibers performed best.

Additionally, predictions of the concrete stresses within the AASHTO ring test have been calculated using Hossain and Weiss’s model [42]. In the absence of accurate methods to directly measure the restrained concrete stress, this method assesses the residual stress development in restrained concrete ring specimens and helps quantify the benefits of adding low volumes of hybrid fibers in reducing concrete-induced stresses. As illustrated in Figure 9, the average concrete stress is calculated based on the average strain measurement on the steel ring and compared to the cracking strain (direct tensile measurement). The results show that the concrete stress did not exceed the cracking capacity of the mix and adding hybrid fibers reduced the concrete stresses, which was also reflected in the cracking area observed on the ring surfaces.

## 5. Conclusions and Discussion

The addition of various combinations of low volume fraction hybrid fiber mix designs was investigated and compared with an HPC control mix. The tests performed on each mix included compressive strength, splitting tensile strength, modulus of elasticity, modulus of rupture, surface resistivity, rapid chloride permeability, free shrinkage, and restrained shrinkage. The following summarizes the results of this experiment:All fiber-reinforced concrete mixes tested display higher compressive strength at early ages, possibly due to the fibers intercepting the microcracks as they occur during loading. As the age of the specimens rises, this increase becomes negligible. This suggests that macro fibers tend to increase early-age compressive strength when compared to other types of fibers.The incorporation of micro and macro polypropylene fiber blend improves the splitting tensile strength over single fiber addition. M16N03 and M32N03 both provide significant tensile strength increases, 18.3% and 8.7%, respectively, over M32, indicating that the hybrid combination of macro and micro polypropylene fibers provides synergy, improving the tensile strength.Hybridizing either polypropylene fiber (micro or macro) with steel fibers resulted in the greatest improvements in modulus of rupture. However, for improvements in shrinkage and permeability properties, combining macro and micro polypropylene fibers is preferred.Restrained shrinkage strain decreases by applying the concept of fiber hybridization; in turn, cracking resistance under restrained conditions also improves. M32N0S5 provided the least number of cracks at 56 days, reducing the number of cracks by 14.2% and 16.7%, respectively, in comparison to monofiber mixes S32 and M32.Restrained shrinkage testing concluded that M32N03 outperformed M32 and S32 by 16.6% and 15.7%, respectively. Micro polypropylene fiber combined with either of the macro fibers (polypropylene or steel) tested enhances the crack resistance of HFHPC, where the differences in fibers sizes and types enhance the cracking resistance at different stages of loading or exposure.

## Figures and Tables

**Figure 1 polymers-14-04934-f001:**
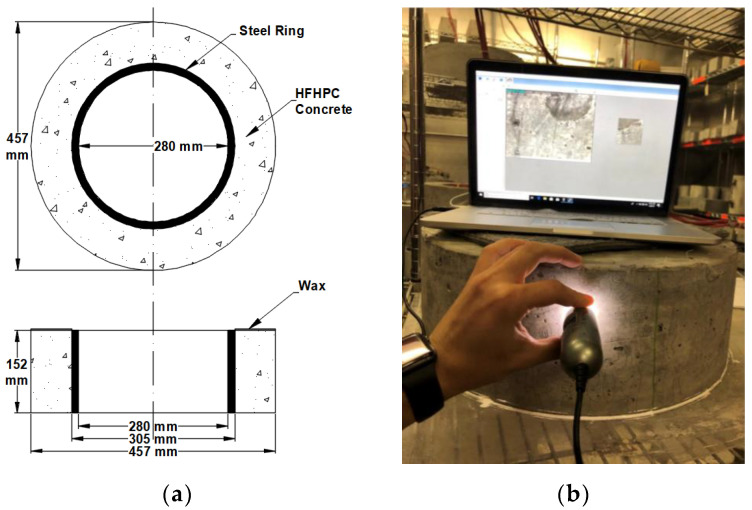
AASHTO PP34 ring test (**a**) dimensions and (**b**) crack mapping process.

**Figure 2 polymers-14-04934-f002:**
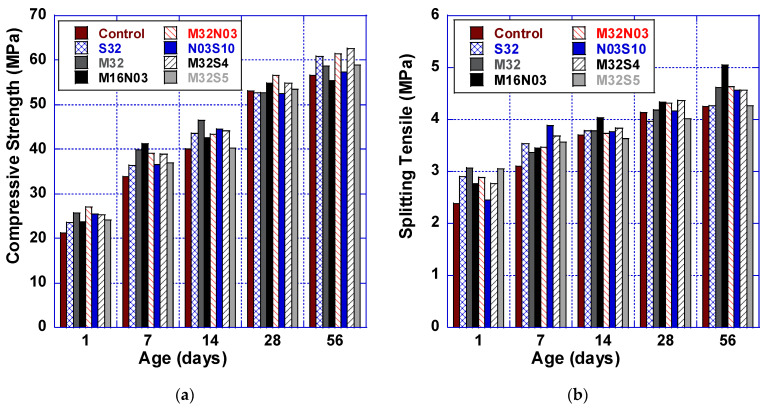
Mechanical properties: (**a**) compressive strength and (**b**) splitting tensile strength.

**Figure 3 polymers-14-04934-f003:**
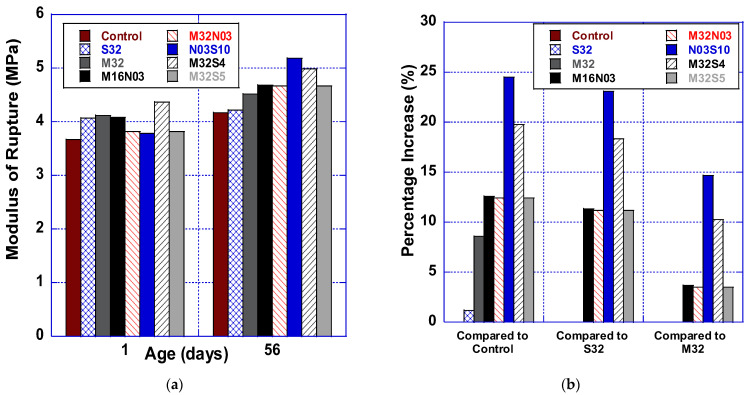
Modulus of rupture (**a**) testing results and (**b**) percentage increase compared to control mixes (Control, S32, M32).

**Figure 4 polymers-14-04934-f004:**
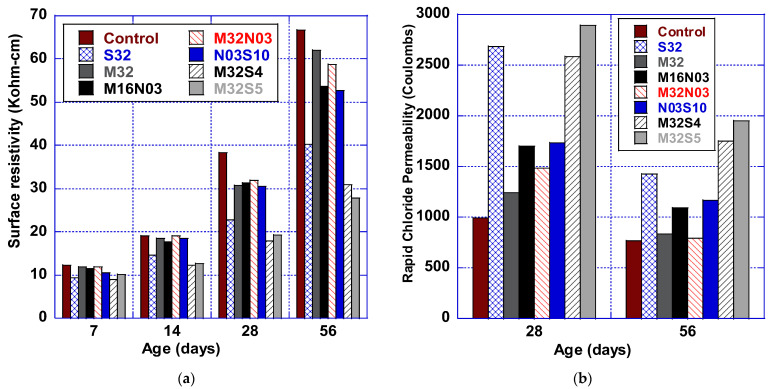
Durability performance of HFHPC: (**a**) surface resistivity (kΩ-cm) and (**b**) rapid chloride permeability (Coulombs).

**Figure 5 polymers-14-04934-f005:**
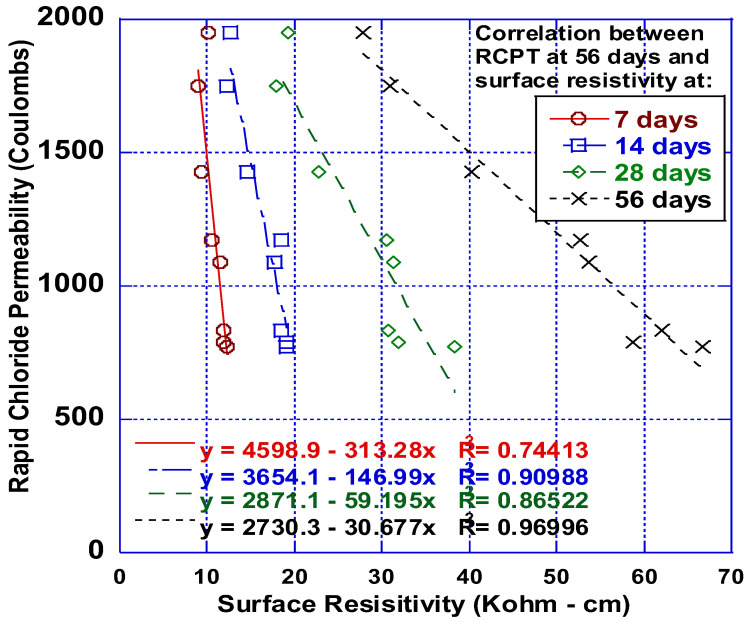
Correlation between the chloride penetrability and surface resistivity.

**Figure 6 polymers-14-04934-f006:**
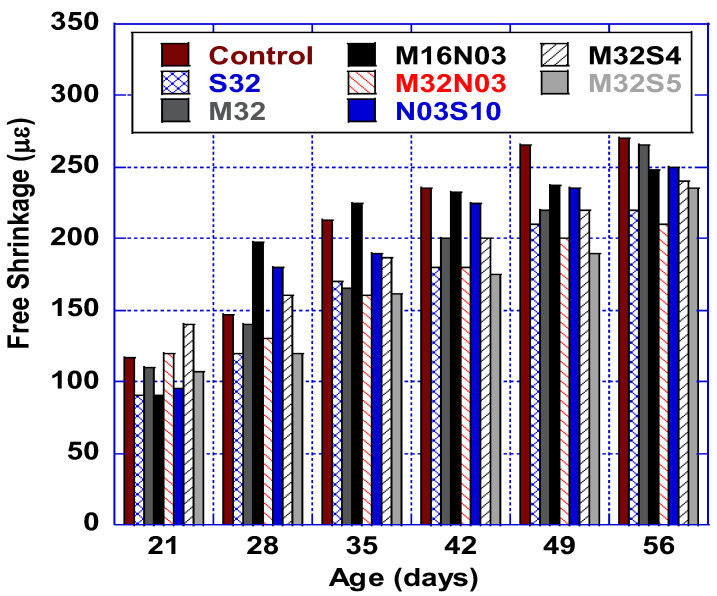
Free shrinkage results.

**Figure 7 polymers-14-04934-f007:**
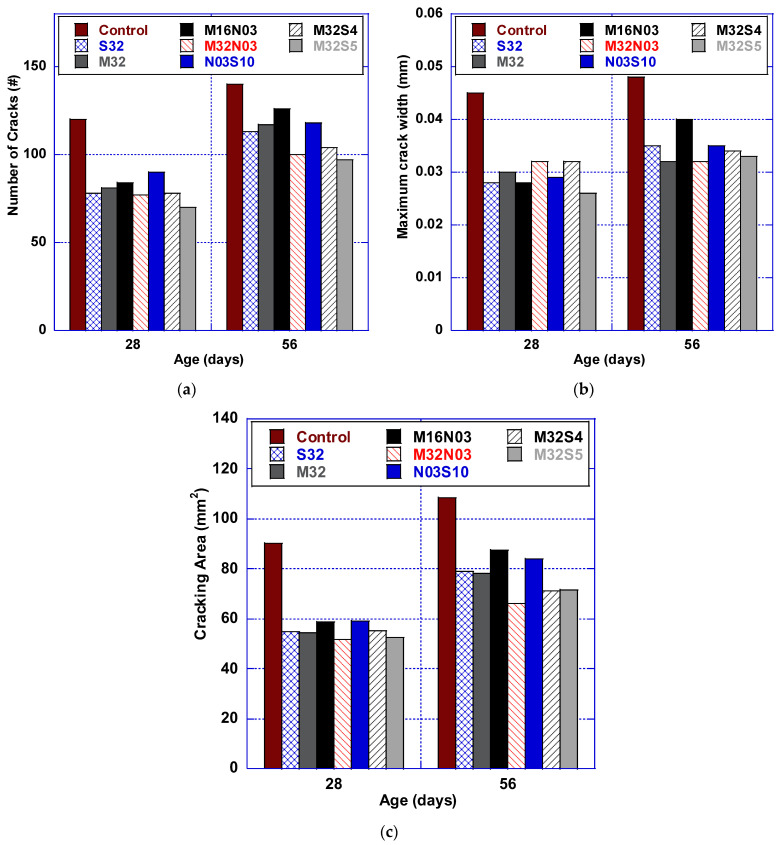
Cracking performance results: (**a**) number of cracks, (**b**) crack width, and (**c**) cracking area.

**Figure 8 polymers-14-04934-f008:**
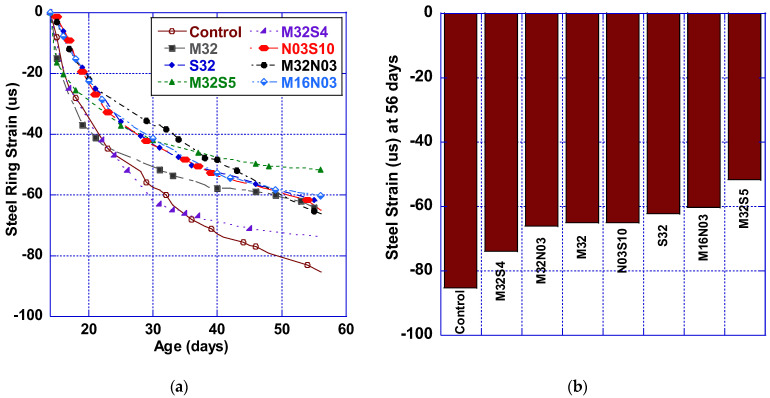
Steel ring strain: (**a**) 14–56 days and (**b**) 56 days.

**Figure 9 polymers-14-04934-f009:**
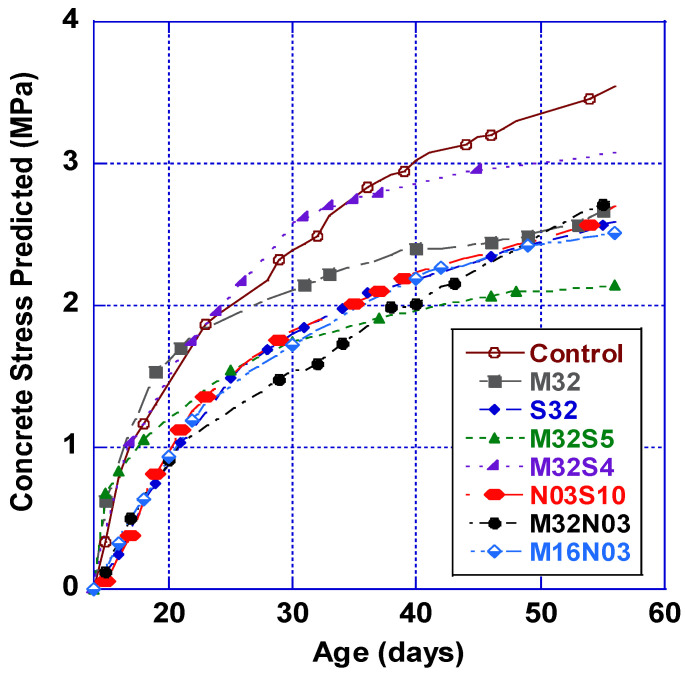
Prediction of the concrete stress using Hossain and Weiss’s prediction model [42].

**Table 1 polymers-14-04934-t001:** Properties of fibers utilized in this study.

Designation	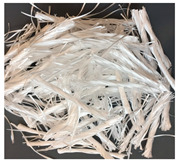	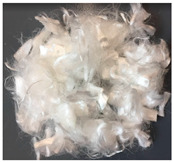	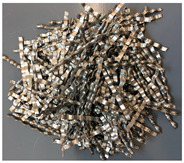
Fiber Type	Polypropylene (M)	Polypropylene (N)	Steel (S)
Comply with	ASTM C1116	ASTM C1116	ASTM C1116
Specific gravity	0.92	0.91	NA
Length, mm	51	19	38
Melting Point, °C	160	160	NA
Tensile strength, MPa	600–650	NA	966–1242
Aspect Ratio	74	NA	34

**Table 2 polymers-14-04934-t002:** Proportioning of the HPC nonfibrous mixture (unit: kg/m^3^).

Mixtures	Cement	FA	SF	Total Cementitious	Sand	Coarse Aggregate		Water	AEA	HRWR
						#67	#8			
HPC	308	77	15	400	660	900	178	152.8	0.08	2.9

SF: silica fume, FA: fly ash, AEA: air-entraining agent.

**Table 3 polymers-14-04934-t003:** Fiber proportions for each mixture.

Mixtures Designation	Fiber Dosage (kg/m^3^)				Fiber Volume Fraction (%)			
	M	N	S	Total	M	N	S	Total
S32	-	-	25	25	-	-	0.32	0.32
M32	3	-	-	3	-	-	0.32	0.32
M16N3	1.5	0.3	-	1.8	0.16	0.03	-	0.19
M32N3	3	3	-	6	0.32	0.03	-	0.35
N3S10	-	0.3	7.4	7.7	-	0.03	0.1	0.13
M32S4	3	-	3	6	0.32	-	0.04	0.36
M32S5	3	-	4	7	0.32	-	0.05	0.37

**Table 4 polymers-14-04934-t004:** Summary of the tests performed for each mix.

Testing	Performance Characteristic	Standard	Size (No. of Sample)
Fresh Properties	Slump	ASTM C143	-
	Air content	ASTM C231	-
Mechanical Strength	Compressive strength	ASTM C39	Cylinder 101 × 203 mm
	Tensile strength	ASTM C496	Cylinder 101 × 203 mm
	Modulus of elasticity	ASTM C469	Cylinder 101 × 203 mm
	Modulus of rupture	ASTM C78	Prism 76 × 76 × 279 mm
Shrinkage Properties	Free shrinkage	ASTM C157	Prism 76 × 76 × 279 mm
	Restrained shrinkage	AASHTO PP34	AASHTO Ring
Durability Testing	Surface resistivity	AASHTO T358	Cylinder 101 × 203 mm
	Rapid chloride permeability	AASHTO T277	Cylinder 101 × 203 mm

**Table 5 polymers-14-04934-t005:** Concrete fresh properties.

Criteria	Control	S32	M32	M16N03	M32N03	N03S10	M32S04	M32S05
HRWR (mL/kg)	2.9	2.6 (−10%)	3.5 (+21%)	3.5 (+21%)	3.2 (+10%)	3.2 (+10%)	3.2 (+10%)	3.2 (+10%)
Slump (cm)	12.7	13.97	8.89	12.7	10.16	17.78	13.97	8.89
Air content (%)	4.5	4.5	4	5	4.8	4.5	4.5	4

**Table 6 polymers-14-04934-t006:** Chloride penetrability classification.

Chloride Permeability	AASHTO TP 95 (Kohm-cm)	ASTM C1202 (Coulombs)
High	Less than 12	More than 4000
Moderate	12–21	2000–4000
Low	21–37	1000–2000
Very low	37–254	100–1000
Negligible	More than 254	Less than 100

**Table 7 polymers-14-04934-t007:** Summary of the cracking performance.

Mix	Age (day)	Number of Cracks	Crack Area (mm^2^)	Max Crack width (mm)
Control	28	120	90.42	0.0394
	56	140	108.45	0.0420
M32	28	81 (−33%)	54.32 (−40%)	0.0290 (−26%)
	56	117 (−16%)	78.26 (−28%)	0.0315 (−25%)
S32	28	78 (−35%)	54.94 (−39%)	0.0274 (−30%)
	56	113 (−19%)	79.10 (−27%)	0.0315 (−25%)
M32N03	28	77 (−36%)	51.61 (−43%)	0.0289 (−26%)
	56	100 (−29%)	65.97 (−39%)	0.0314 (−25%)
M32S4	28	78 (−35%)	55.26 (−39%)	0.0315 (−20%)
	56	104 (−26%)	71.06 (−34%)	0.0350 (−19%)
M32S5	28	70 (−42%)	52.42 (−42%)	0.0259 (−34%)
	56	97 (−31%)	71.45 (−34%)	0.0325 (−23%)
M16N03	28	84 (−30%)	58.58 (−35%)	0.0264 (−33%)
	56	126 (−10%)	87.64 (−19%)	0.0359 (−15%)
N03S10	28	90 (−25%)	59.29 (−34%)	0.0279 (−29%)
	56	118 (−16%)	84.16 (−22%)	0.0351 (−17%)

## Data Availability

The data presented in this study are available on request from the corresponding author.

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
