# Peer review of "Restrained Shrinkage of High-Performance Ready-Mix Concrete Reinforced with Low Volume Fraction of Hybrid Fibers"

_polymers, 2022, doi:10.3390/polym14224934_

Round 1
Reviewer 1 Report
In present work, mechanical and durability of concrete reinforced by low-volume-fraction hybridizing fibers were investigated. The experimental and results were well designed and displayed. Minor revision is suggested.
1. Page 2, Line 81. ‘the external restraint is due to’ capitalize the first letter.
2. The basis for selection of hybridizing ratio between M, N or S fiber should be described in the mixture design.
3. It is suggested that the authors can compare your results with references concerning high volume fraction of hybridized fibers. This may help to confirm whether reducing fiber volume content can help to reduce the negative effect on permeability.
4. The AASHTO PP34 ring test schematic diagram in Fig. 6 should be placed in Section 2.3.
5. The minimum crack width that can be recognized by the equipment in Fig. 6(b) should be specified.
6. Are the number, width and area of cracks automatically identified and counted by the equipment or manually recorded?
Author Response
Thank you for your time and for reviewing our paper, please find the attached answeres and the attached modified manuscript.
Please let us know in case any other modifications are required, Best

Reviewer 2 Report
Dear Authors,
In the article there is presented a very interesting problem of used hybrid fibers on the fresh, physical and mechanical properties of concrete. To make the content of the article more clear to a reader authors should take into consideration the following points:
Why have not been tested the samples as fiber concrete? Exist norm for test concrete with fibre. Why have not been tested residual strength? This is one of the basic properties of FRC.
How many compressive strength, tensile strength, modulus of elasticity, modulus of rupture and e.t.c specimen were used? For example, how many specimens used to consider compressive strength after 7th day?
What is the standard deviation for average compressive strength, tensile strength, modulus of elasticity and e.t.c (figure 1, 2, 3, 5, 7)? How big is the dispersion of results?
There are no correlation coefficients (r) in Figure 8 and 9. What is the function of Steel Ring Strain in different concrete. Why the research was discontinued on day 56. The values on the chart keep growing ...
Please supplement the literature about study on fibre reinforced fine aggregate concrete based on waste sand. Especially, the effect of small amount of fibers in concrete based on waste sand, on the shrinkage strains.Author Response
Thank you for your time and for reviewing our paper, please find the attached answeres and the attached modified manuscript.
Please let us know in case any other modifications are required, Best
